# Optimal Testing for Properties of Distributions

**Jayadev Acharya, Constantinos Daskalakis, Gautam Kamath**
EECS, MIT
{jayadev, costis, g}@mit.edu

## Abstract

Given samples from an unknown discrete distribution $p$, is it possible to distinguish whether $p$ belongs to some class of distributions $\mathcal{C}$ versus $p$ being far from every distribution in $\mathcal{C}$? This fundamental question has received tremendous attention in statistics, focusing primarily on asymptotic analysis, as well as in information theory and theoretical computer science, where the emphasis has been on small sample size and computational complexity. Nevertheless, even for basic properties of discrete distributions such as monotonicity, independence, log-concavity, unimodality, and monotone-hazard rate, the optimal sample complexity is unknown.

We provide a general approach via which we obtain sample-optimal and computationally efficient testers for all these distribution families. At the core of our approach is an algorithm which solves the following problem: Given samples from an unknown distribution $p$, and a known distribution $q$, are $p$ and $q$ close in $\chi^2$-distance, or far in total variation distance?

The optimality of our testers is established by providing matching lower bounds, up to constant factors. Finally, a necessary building block for our testers and an important byproduct of our work are the first known computationally efficient proper learners for discrete log-concave, monotone hazard rate distributions.

## 1 Introduction

The quintessential scientific question is whether an unknown object has some property, i.e. whether a model from a specific class fits the object's observed behavior. If the unknown object is a probability distribution, $p$, to which we have sample access, we are typically asked to distinguish whether $p$ belongs to some class $\mathcal{C}$ or whether it is sufficiently far from it.

This question has received tremendous attention in the field of statistics (see, e.g., [1, 2]), where test statistics for important properties such as the ones we consider here have been proposed. Nevertheless, the emphasis has been on asymptotic analysis, characterizing the rates of convergence of test statistics under null hypotheses, as the number of samples tends to infinity. In contrast, we wish to study the following problem in the small sample regime:

> $\Pi(\mathcal{C}, \varepsilon)$: Given a family of distributions $\mathcal{C}$, some $\varepsilon > 0$, and sample access to an unknown distribution $p$ over a discrete support, how many samples are required to distinguish between $p \in \mathcal{C}$ versus $d_{\mathrm{TV}}(p, \mathcal{C}) > \varepsilon$?[1]

The problem has been studied intensely in the literature on property testing and sublinear algorithms [3, 4, 5], where the emphasis has been on characterizing the optimal tradeoff between $p$'s support size and the accuracy $\varepsilon$ in the number of samples. Several results have been obtained, roughly

clustering into three groups, where (i) $\mathcal{C}$ is the class of monotone distributions over $[n]$, or more generally a poset [6, 7]; (ii) $\mathcal{C}$ is the class of independent, or $k$-wise independent distributions over a hypergrid [8, 9]; and (iii) $\mathcal{C}$ contains a single-distribution $q$, and the problem becomes that of testing whether $p$ equals $q$ or is far from it [8, 10, 11, 13].

With respect to (iii), [13] exactly characterizes the number of samples required to test identity to each distribution $q$, providing a single tester matching this bound simultaneously for all $q$. Nevertheless, this tester and its precursors are not applicable to the composite identity testing problem that we consider. If our class $\mathcal{C}$ were finite, we could test against each element in the class, albeit this would not necessarily be sample optimal. If our class $\mathcal{C}$ were a continuum, we would need *tolerant* identity testers, which tend to be more expensive in terms of sample complexity [12], and result in substantially suboptimal testers for the classes we consider. Or we could use approaches related to generalized likelihood ratio test, but their behavior is not well-understood in our regime, and optimizing likelihood over our classes becomes computationally intense.

**Our Contributions**   We obtain sample-optimal and computationally efficient testers for $\Pi(\mathcal{C}, \varepsilon)$ for the most fundamental shape restrictions to a distribution. Our contributions are the following:

1. For a known distribution $q$ over $[n]$, and sample access to $p$, we show that distinguishing the cases: $(a)$ whether the $\chi^2$-distance between $p$ and $q$ is at most $\varepsilon^2/2$, versus $(b)$ the $\ell_1$ distance between $p$ and $q$ is at least $2\varepsilon$, requires $\Theta(\sqrt{n}/\varepsilon^2)$ samples. As a corollary, we obtain an alternate argument that shows that identity testing requires $\Theta(\sqrt{n}/\varepsilon^2)$ samples (previously shown in [13]).

2. For the class $\mathcal{C} = \mathcal{M}_n^d$ of monotone distributions over $[n]^d$ we require an optimal $\Theta\left(n^{d/2}/\varepsilon^2\right)$ number of samples, where prior work requires $\Omega\left(\sqrt{n}\log n/\varepsilon^6\right)$ samples for $d = 1$ and $\tilde{\Omega}\left(n^{d-1/2}\text{poly}\left(1/\varepsilon\right)\right)$ for $d > 1$ [6, 7]. Our results improve the exponent of $n$ with respect to $d$, shave all logarithmic factors in $n$, and improve the exponent of $\varepsilon$ by at least a factor of 2.

    (a) A useful building block and interesting byproduct of our analysis is extending Birgé's oblivious decomposition for single-dimensional monotone distributions [14] to monotone distributions in $d \geq 1$, and to the stronger notion of $\chi^2$-distance. See Section C.1.

    (b) Moreover, we show that $O(\log^d n)$ samples suffice to learn a monotone distribution over $[n]^d$ in $\chi^2$-distance. See Lemma 3 for the precise statement.

3. For the class $\mathcal{C} = \Pi_d$ of product distributions over $[n_1] \times \cdots \times [n_d]$, our algorithm requires $O\left(\left(\left(\prod_\ell n_\ell\right)^{1/2} + \sum_\ell n_\ell\right)/\varepsilon^2\right)$ samples. We note that a product distribution is one where all marginals are independent, so this is equivalent to testing if a collection of random variables are all independent. In the case where $n_\ell$'s are large, then the first term dominates, and the sample complexity is $O((\prod_\ell n_\ell)^{1/2}/\varepsilon^2)$. In particular, when $d$ is a constant and all $n_\ell$'s are equal to $n$, we achieve the optimal sample complexity of $\Theta(n^{d/2}/\varepsilon^2)$. To the best of our knowledge, this is the first result for $d \geq 3$, and when $d = 2$, this improves the previously known complexity from $O\left(\frac{n}{\varepsilon^6}\text{polylog}(n/\varepsilon)\right)$ [8, 15], significantly improving the dependence on $\varepsilon$ and shaving all logarithmic factors.

4. For the classes $\mathcal{C} = \mathcal{LCD}_n$, $\mathcal{C} = \mathcal{MHR}_n$ and $\mathcal{C} = \mathcal{U}_n$ of log-concave, monotone-hazard-rate and unimodal distributions over $[n]$, we require an optimal $\Theta\left(\sqrt{n}/\varepsilon^2\right)$ number of samples. Our testers for $\mathcal{LCD}_n$ and $\mathcal{C} = \mathcal{MHR}_n$ are to our knowledge the first for these classes for the low sample regime we are studying—see [16] and its references for statistics literature on the asymptotic regime. Our tester for $\mathcal{U}_n$ improves the dependence of the sample complexity on $\varepsilon$ by at least a factor of 2 in the exponent, and shaves all logarithmic factors in $n$, compared to testers based on testing monotonicity.

    (a) A useful building block and important byproduct of our analysis are the first computationally efficient algorithms for properly learning log-concave and monotone-hazard-rate distributions, to within $\varepsilon$ in total variation distance, from $\text{poly}(1/\varepsilon)$ samples, independent of the domain size $n$. See Corollaries 4 and 6. Again, these are the first computationally efficient algorithms to our knowledge in the low sample regime. [17] provide algorithms for density estimation, which are non-proper, i.e. will approximate an unknown distribution from these classes with a distribution that does not belong to these classes. On the other hand, the statistics literature focuses on maximum-likelihood estimation in the asymptotic regime—see e.g. [18] and its references.

5. For all the above classes we obtain matching lower bounds, showing that the sample complexity of our testers is optimal with respect to $n$, $\varepsilon$ and when applicable $d$. See Section 8. Our lower bounds are based on extending Paninski's lower bound for testing uniformity [10].

**Our Techniques**    At the heart of our tester lies a novel use of the $\chi^2$ statistic. Naturally, the $\chi^2$ and its related $\ell_2$ statistic have been used in several of the afore-cited results. We propose a new use of the $\chi^2$ statistic enabling our optimal sample complexity. The essence of our approach is to first draw a small number of samples (independent of $n$ for log-concave and monotone-hazard-rate distributions and only logarithmic in $n$ for monotone and unimodal distributions) to approximate the unknown distribution $p$ in $\chi^2$ distance. If $p \in \mathcal{C}$, our learner is required to output a distribution $q$ that is $O(\varepsilon)$-close to $\mathcal{C}$ in total variation and $O(\varepsilon^2)$-close to $p$ in $\chi^2$ distance. Then some analysis reduces our testing problem to distinguishing the following cases:

- $p$ and $q$ are $O(\varepsilon^2)$-close in $\chi^2$ distance; this case corresponds to $p \in \mathcal{C}$.
- $p$ and $q$ are $\Omega(\varepsilon)$-far in total variation distance; this case corresponds to $d_{\mathrm{TV}}(p, \mathcal{C}) > \varepsilon$.

We draw a comparison with *robust identity testing*, in which one must distinguish whether $p$ and $q$ are $c_1\varepsilon$-close or $c_2\varepsilon$-far in total variation distance, for constants $c_2 > c_1 > 0$. In [12], Valiant and Valiant show that $\Omega(n/\log n)$ samples are required for this problem – a nearly-linear sample complexity, which may be prohibitively large in many settings. In comparison, the problem we study tests for $\chi^2$ closeness rather than total variation closeness: a relaxation of the previous problem. However, our tester demonstrates that this relaxation allows us to achieve a substantially sublinear complexity of $O(\sqrt{n}/\varepsilon^2)$. On the other hand, this relaxation is still tight enough to be useful, demonstrated by our application in obtaining sample-optimal testers.

We note that while the $\chi^2$ statistic for testing hypothesis is prevalent in statistics providing optimal error exponents in the large-sample regime, to the best of our knowledge, in the small-sample regime, *modified-versions* of the $\chi^2$ statistic have only been recently used for *closeness-testing* in [19, 20, 21] and for testing uniformity of monotone distributions in [22]. In particular, [19] design an unbiased statistic for estimating the $\chi^2$ distance between two *unknown* distributions.

**Organization**    In Section 4, we show that a version of the $\chi^2$ statistic, appropriately excluding certain elements of the support, is sufficiently well-concentrated to distinguish between the above cases. Moreover, the sample complexity of our algorithm is optimal for most classes. Our base tester is combined with the afore-mentioned extension of Birgé's decomposition theorem to test monotone distributions in Section 5 (see Theorem 2 and Corollary 1), and is also used to test independence of distributions in Section 6 (see Theorem 3).

In Section 7, we give our results on testing unimodal, log-concave and monotone hazard rate distributions. Naturally, there are several bells and whistles that we need to add to the above skeleton to accommodate all classes of distributions that we are considering. In Remark 1 we mention the additional modifications for these classes.

**Related Work.**    For the problems that we study in this paper, we have provided the related works in the previous section along with our contributions. We cannot do justice to the role of shape restrictions of probability distributions in probabilistic modeling and testing. It suffices to say that the classes of distributions that we study are fundamental, motivating extensive literature on their learning and testing [23]. In the recent times, there has been work on shape restricted statistics, pioneered by Jon Wellner, and others. [24, 25] study estimation of monotone and $k$-monotone densities, and [26, 27] study estimation of log-concave distributions. Due to the sheer volume of literature in statistics in this field, we will restrict ourselves to those already referenced.

As we have mentioned, statistics has focused on the asymptotic regime as the number of samples tends to infinity. Instead we are considering the low sample regime and are more stringent about the behavior of our testers, requiring 2-sided guarantees. We want to accept if the unknown distribution is in our class of interest, and also reject if it is far from the class. For this problem, as discussed above, there are few results when $\mathcal{C}$ is a whole class of distributions. Closer related to our paper is the line of papers [6, 7, 28] for monotonicity testing, albeit these papers have sub-optimal sample complexity as discussed above. Testing independence of random variables has a long history in statisics [29, 30]. The theoretical computer science community has also considered the problem of

testing independence of two random variables [8, 15]. While our results sharpen the case where the variables are over domains of equal size, they demonstrate an interesting asymmetric upper bound when this is not the case. More recently, Acharya and Daskalakis provide optimal testers for the family of Poisson Binomial Distributions [31].

Finally, contemporaneous work of Canonne et al [32] provides a generic algorithm and lower bounds for the single-dimensional families of distributions considered here. We note that their algorithm has a sample complexity which is suboptimal in both $n$ and $\varepsilon$, while our algorithms are optimal. Their algorithm also extends to mixtures of these classes, though some of these extensions are not computationally efficient. They also provide a framework for proving lower bounds, giving the optimal bounds for many classes when $\varepsilon$ is sufficiently large with respect to $1/n$. In comparison, we provide these lower bounds unconditionally by modifying Paninski's construction [10] to suit the classes we consider.

## 2 Preliminaries

We use the following probability distances in our paper.

The *total variation distance* between distributions $p$ and $q$ is $d_{\mathrm{TV}}(p,q) \overset{\mathrm{def}}{=} \sup_A |p(A) - q(A)| = \frac{1}{2}\|p - q\|_1$. The $\chi^2$-*distance* between $p$ and $q$ over $[n]$ is defined as $\chi^2(p,q) \overset{\mathrm{def}}{=} \sum_{i \in [n]} \frac{(p_i - q_i)^2}{q_i}$. The *Kolmogorov distance* between two probability measures $p$ and $q$ over an ordered set (*e.g.*, $\mathbf{R}$) with cumulative density functions $F_p$ and $F_q$ is $d_{\mathrm{K}}(p,q) \overset{\mathrm{def}}{=} \sup_{x \in \mathbb{R}} |F_p(x) - F_q(x)|$.

Our paper is primarily concerned with testing against classes of distributions, defined formally as:

**Definition 1.** *Given $\varepsilon \in (0,1]$ and sample access to a distribution $p$, an algorithm is said to* test *a class $\mathcal{C}$ if it has the following guarantees:*

- *If $p \in \mathcal{C}$, the algorithm outputs* ACCEPT *with probability at least $2/3$;*

- *If $d_{\mathrm{TV}}(p, \mathcal{C}) \geq \varepsilon$, the algorithm outputs* REJECT *with probability at least $2/3$.*

We note the following useful relationships between these distances [33]:

**Proposition 1.** $d_{\mathrm{K}}(p,q)^2 \leq d_{\mathrm{TV}}(p,q)^2 \leq \frac{1}{4}\chi^2(p,q)$.

**Definition 2.** *An $\eta$-effective support of a distribution $p$ is any set $S$ such that $p(S) \geq 1 - \eta$.*

The *flattening* of a function $f$ over a subset $S$ is the function $\bar{f}$ such that $\bar{f}_i = p(S)/|S|$.

**Definition 3.** *Let $p$ be a distribution, and support $I_1, \ldots$ is a partition of the domain. The flattening of $p$ with respect to $I_1, \ldots$ is the distribution $\bar{p}$ which is the flattening of $p$ over the intervals $I_1, \ldots$.*

**Poisson Sampling** Throughout this paper, we use the standard Poissonization approach. Instead of drawing exactly $m$ samples from a distribution $p$, we first draw $m' \sim \mathrm{Poisson(m)}$, and then draw $m'$ samples from $p$. As a result, the number of times different elements in the support of $p$ occur in the sample become independent, giving much simpler analyses. In particular, the number of times we will observe domain element $i$ will be distributed as $\mathrm{Poisson(mp_i)}$, independently for each $i$. Since $\mathrm{Poisson(m)}$ is tightly concentrated around $m$, this additional flexibility comes only at a sub-constant cost in the sample complexity with an inversely exponential in $m$, additive increase in the error probability.

## 3 The Testing Algorithm – An Overview

Our algorithm for testing a class $\mathcal{C}$ can be decomposed into three steps.

**Near-proper learning in $\chi^2$-distance.** Our first step requires learning with very specific guarantees. Given sample access to $p \in \mathcal{C}$, we wish to output $q$ such that (i) $q$ is *close* to $\mathcal{C}$ in total variation distance, and (ii) $p$ and $q$ are $O(\varepsilon^2)$-close in $\chi^2$-distance on an $\varepsilon$-effective support[2] of $p$. When

$p$ is not in $\mathcal{C}$, we do not guarantee anything about $q$. From an information theoretic standpoint, this problem is harder than learning the distribution in total variation, since $\chi^2$-distance is more restrictive than total variation distance. Nonetheless, for the structured classes we consider, we are able to learn in $\chi^2$ by modifying the approaches to learn in total variation.

**Computation of distance to class.** The next step is to see if the hypothesis $q$ is close to the class $\mathcal{C}$ or not. Since we have an explicit description of $q$, this step requires no further samples from $p$, i.e. it is purely computational. If we find that $q$ is far from the class $\mathcal{C}$, then it must be that $p \notin \mathcal{C}$, as otherwise the guarantees from the previous step would imply that $q$ is close to $\mathcal{C}$. Thus, if it is not, we can terminate the algorithm at this point.

$\chi^2$-**testing.** At this point, the previous two steps guarantee that our distribution $q$ is such that:

- If $p \in \mathcal{C}$, then $p$ and $q$ are close in $\chi^2$ distance on a (known) effective support of $p$;

- If $d_{\mathrm{TV}}(p, \mathcal{C}) \geq \varepsilon$, then $p$ and $q$ are far in total variation distance.

We can distinguish between these two cases using $O(\sqrt{n}/\varepsilon^2)$ samples with a simple statistical $\chi^2$-test, that we describe in Section 4.

Using the above three-step approach, our tester, as described in the next section, can directly test monotonicity, log-concavity, and monotone hazard rate. With an extra trick, using Kolmogorov's max inequality, it can also test unimodality.

# 4  A Robust $\chi^2$-$\ell_1$ Identity Test

Our main result in this section is Theorem 1.

**Theorem 1.** *Given $\varepsilon \in (0, 1]$, a class of probability distributions $\mathcal{C}$, sample access to a distribution $p$, and an explicit description of a distribution $q$, both over $[n]$ with the following properties:*

> ***Property 1.*** *$d_{\mathrm{TV}}(q, \mathcal{C}) \leq \frac{\varepsilon}{2}$.*

> ***Property 2.*** *If $p \in \mathcal{C}$, then $\chi^2(p, q) \leq \frac{\varepsilon^2}{500}$.*

*Then there exists an algorithm such that: If $p \in \mathcal{C}$, it outputs* ACCEPT *with probability at least $2/3$; If $d_{\mathrm{TV}}(p, \mathcal{C}) \geq \varepsilon$, it outputs* REJECT *with probability at least $2/3$. The time and sample complexity of this algorithm are $O\left(\sqrt{n}/\varepsilon^2\right)$.*

*Proof.* Algorithm 1 describes a $\chi^2$ testing procedure that gives the guarantee of the theorem.

---
**Algorithm 1** Chi-squared testing algorithm
---
1: **Input:** $\varepsilon$; an explicit distribution $q$; (Poisson) $m$ samples from a distribution $p$, where $N_i$ denotes the number of occurrences of the $i$th domain element.
2: $\mathcal{A} \leftarrow \{i : q_i \geq \varepsilon^2/50n\}$
3: $Z \leftarrow \sum_{i \in \mathcal{A}} \frac{(N_i - mq_i)^2 - N_i}{mq_i}$
4: **if** $Z \leq m\varepsilon^2/10$ **return** close
5: **else return** far
---

In Section A we compute the mean and variance of the statistic $Z$ (defined in Algorithm 1) as:

$$\mathbb{E}\left[Z\right] = m \cdot \sum_{i \in \mathcal{A}} \frac{(p_i - q_i)^2}{q_i} = m \cdot \chi^2(p_\mathcal{A}, q_\mathcal{A}), \ \ \mathrm{Var}\left[Z\right] = \sum_{i \in \mathcal{A}} \left[2\frac{p_i^2}{q_i^2} + 4m \cdot \frac{p_i \cdot (p_i - q_i)^2}{q_i^2}\right] \quad (1)$$

where by $p_\mathcal{A}$ and $q_\mathcal{A}$ we denote respectively the vectors $p$ and $q$ restricted to the coordinates in $\mathcal{A}$, and we slightly abuse notation when we write $\chi^2(p_\mathcal{A}, q_\mathcal{A})$, as these do not then correspond to probability distributions.

Lemma 1 demonstrates the separation in the means of the statistic $Z$ in the two cases of interest, *i.e.*, $p \in \mathcal{C}$ versus $d_{\mathrm{TV}}(p, \mathcal{C}) \geq \varepsilon$, and Lemma 2 shows the separation in the variances in the two cases. These two results are proved in Section B.

**Lemma 1.** *If $p \in \mathcal{C}$, then $\mathbb{E}[Z] \leq m\varepsilon^2/500$. If $d_{\mathrm{TV}}(p, \mathcal{C}) \geq \varepsilon$, then $\mathbb{E}[Z] \geq m\varepsilon^2/5$.*

**Lemma 2.** *Let $m \geq 20000\sqrt{n}/\varepsilon^2$. If $p \in \mathcal{C}$ then $\mathrm{Var}[Z] \leq \frac{1}{500000}m^2\varepsilon^4$. If $d_{\mathrm{TV}}(p, \mathcal{C}) \geq \varepsilon$, then $\mathrm{Var}[Z] \leq \frac{1}{100}E[Z]^2$.*

Assuming Lemmas 1 and 2, Theorem 1 is now a simple application of Chebyshev's inequality.

When $p \in \mathcal{C}$, we have that $\mathbb{E}[Z] + \sqrt{3\,\mathrm{Var}[Z]} \leq \left(1/500 + \sqrt{3/500000}\right)m\varepsilon^2 \leq m\varepsilon^2/200$. Thus, Chebyshev's inequality gives

$$\Pr\left[Z \geq m\varepsilon^2/10\right] \leq \Pr\left[Z \geq m\varepsilon^2/200\right] \leq \Pr\left[Z - \mathbb{E}[Z] \geq \sqrt{3}\,\mathrm{Var}[Z]^{1/2}\right] \leq 1/3.$$

When $d_{\mathrm{TV}}(p, \mathcal{C}) \geq \varepsilon$, $\mathbb{E}[Z] - \sqrt{3\,\mathrm{Var}[Z]} \geq \left(1 - \sqrt{3/100}\right)E[Z] \geq 3m\varepsilon^2/20$. Therefore,

$$\Pr\left[Z \leq m\varepsilon^2/10\right] \leq \Pr\left[Z \leq 3m\varepsilon^2/20\right] \leq \Pr\left[Z - \mathbb{E}[Z] \leq -\sqrt{3}\,\mathrm{Var}[Z]^{1/2}\right] \leq 1/3. \quad \square$$

This proves the correctness of Algorithm 1. For the running time, we divide the summation in $Z$ into the elements for which $N_i > 0$ and $N_i = 0$. When $N_i = 0$, the contribution of the term to the summation is $mq_i$, and we can sum them up by subtracting the total probability of all elements appearing at least once from 1.

**Remark 1.** *To apply Theorem 1, we need to learn distribution in $\mathcal{C}$ and find a $q$ that is $O(\varepsilon^2)$-close in $\chi^2$-distance to $p$. For the class of monotone distributions, we are able to efficiently obtain such a $q$, which immediately implies sample-optimal learning algorithms for this class. However, for some classes, we may not be able to learn a $q$ with such strong guarantees, and we must consider modifications to our base testing algorithm.*

*For example, for log-concave and monotone hazard rate distributions, we can obtain a distribution $q$ and a set $S$ with the following guarantees:*

*- If $p \in \mathcal{C}$, then $\chi^2(p_S, q_S) \leq O(\varepsilon^2)$ and $p(S) \geq 1 - O(\varepsilon)$;*

*- If $d_{\mathrm{TV}}(p, \mathcal{C}) \geq \varepsilon$, then $d_{\mathrm{TV}}(p, q) \geq \varepsilon/2$. In this scenario, the tester will simply pretend that the support of $p$ and $q$ is $S$, ignoring any samples and support elements in $[n] \setminus S$. Analysis of this tester is extremely similar to Theorem 1. In particular, we can still show that the statistic $Z$ will be separated in the two cases. When $p \in \mathcal{C}$, excluding $[n] \setminus S$ will only reduce $Z$. On the other hand, when $d_{\mathrm{TV}}(p, \mathcal{C}) \geq \varepsilon$, since $p(S) \geq 1 - O(\varepsilon)$, $p$ and $q$ must still be far on the remaining support, and we can show that $Z$ is still sufficiently large. Therefore, a small modification allows us to handle this case with the same sample complexity of $O(\sqrt{n}/\varepsilon^2)$.*

*For unimodal distributions, we are even unable to identify a large enough subset of the support where the $\chi^2$ approximation is guaranteed to be tight. But we can show that there exists a light enough piece of the support (in terms of probability mass under $p$) that we can exclude to make the $\chi^2$ approximation tight. Given that we only use Chebyshev's inequality to prove the concentration of the test statistic, it would seem that our lack of knowledge of the piece to exclude would involve a union bound and a corresponding increase in the required number of samples. We avoid this through a careful application of Kolmogorov's max inequality in our setting. See Theorem 7 of Section 7.*

## 5  Testing Monotonicity

As the first application of our testing framework, we will demonstrate how to test for monotonicity. Let $d \geq 1$, and $\mathbf{i} = (i_1, \ldots, i_d), \mathbf{j} = (j_1, \ldots, j_d) \in [n]^d$. We say $\mathbf{i} \succcurlyeq \mathbf{j}$ if $i_l \geq j_l$ for $l = 1, \ldots, d$. A distribution $p$ over $[n]^d$ is monotone (decreasing) if for all $\mathbf{i} \succcurlyeq \mathbf{j}$, $p_{\mathbf{i}} \leq p_{\mathbf{j}}$[3].

We follow the steps in the overview. The learning result we show is as follows (proved in Section C).

**Lemma 3.** *Let $d \geq 1$. There is an algorithm that takes $m = O((d\log(n)/\varepsilon^2)^d/\varepsilon^2)$ samples from a distribution $p$ over $[n]^d$, and outputs a distribution $q$ such that if $p$ is monotone, then with probability at least $5/6$, $\chi^2(p,q) \leq \frac{\varepsilon^2}{500}$. Furthermore, the distance of $q$ to monotone distributions can be computed in time $poly(m)$.*

This accomplishes the first two steps in the overview. In particular, if the distance of $q$ from monotone distributions is more than $\varepsilon/2$, we declare that $p$ is not monotone. Therefore, Property 1 in Theorem 1 is satisfied, and the lemma states that Property 2 holds with probability at least $5/6$. We then proceed to the $\chi^2 - \ell_1$ test. At this point, we have precisely the guarantees needed to apply Theorem 1 over $[n]^d$, directly implying our main result of this section:

**Theorem 2.** *For any $d \geq 1$, there exists an algorithm for testing monotonicity over $[n]^d$ with sample complexity*

$$O\left(\frac{n^{d/2}}{\varepsilon^2} + \left(\frac{d\log n}{\varepsilon^2}\right)^d \cdot \frac{1}{\varepsilon^2}\right)$$

*and time complexity $O\left(n^{d/2}/\varepsilon^2 + \mathrm{poly}(\log n, 1/\varepsilon)^d\right)$.*

In particular, this implies the following optimal algorithms for monotonicity testing for all $d \geq 1$:

**Corollary 1.** *Fix any $d \geq 1$, and suppose $\varepsilon > \sqrt{d\log n}/n^{1/4}$. Then there exists an algorithm for testing monotonicity over $[n]^d$ with sample complexity $O\left(n^{d/2}/\varepsilon^2\right)$.*

We note that the class of monotone distributions is the simplest of the classes we consider. We now consider testing for log-concavity, monotone hazard rate, and unimodality, all of which are much more challenging to test. In particular, these classes require a more sophisticated structural understanding, more complex proper $\chi^2$-learning algorithms, and non-trivial modifications to our $\chi^2$-tester. We have already given some details on the required adaptations to the tester in Remark 1.

Our algorithms for learning these classes use convex programming. One of the main challenges is to enforce log-concavity of the PDF when learning $\mathcal{LCD}_n$ (respectively, of the CDF when learning $\mathcal{MHR}_n$), while simultaneously enforcing closeness in total variation distance. This involves a careful choice of our variables, and we exploit structural properties of the classes to ensure the soundness of particular Taylor approximations. We encourage the reader to refer to the proofs of Theorems 7, 8, and 9 for more details.

## 6 Testing Independence of Random Variables

Let $\mathcal{X} \stackrel{\text{def}}{=} [n_1] \times \ldots \times [n_d]$, and let $\Pi_d$ be the class of all product distributions over $\mathcal{X}$. Similar to learning monotone distributions in $\chi^2$ distance we prove the following result in Section E.

**Lemma 4.** *There is an algorithm that takes $O\left((\sum_{\ell=1}^{d} n_\ell)/\varepsilon^2\right)$ samples from a distribution $p$ and outputs a $q \in \Pi_d$ such that if $p \in \Pi_d$, then with probability at least $5/6$, $\chi^2(p,q) \leq O(\varepsilon^2)$.*

The distribution $q$ always satisfies Property 1 since it is in $\Pi_d$, and by this lemma, with probability at least $5/6$ satisfies Property 2 in Theorem 1. Therefore, we obtain the following result.

**Theorem 3.** *For any $d \geq 1$, there exists an algorithm for testing independence of random variables over $[n_1] \times \ldots [n_d]$ with sample and time complexity $O\left(\left((\prod_{\ell=1}^{d} n_\ell)^{1/2} + \sum_{\ell=1}^{d} n_\ell\right)/\varepsilon^2\right)$.*

When $d = 2$ and $n_1 = n_2 = n$ this improves the result of [8] for testing independence of two random variables.

**Corollary 2.** *Testing if two distributions over $[n]$ are independent has sample complexity $\Theta(n/\varepsilon^2)$.*

## 7 Testing Unimodality, Log-Concavity and Monotone Hazard Rate

Unimodal distributions over $[n]$ (denoted by $\mathcal{U}_n$) are all distributions $p$ for which there exists an $i^*$ such that $p_i$ is non-decreasing for $i \leq i^*$ and non-increasing for $i \geq i^*$. Log-concave distributions over $[n]$ (denoted by $\mathcal{LCD}_n$), is the sub-class of unimodal distributions for which $p_{i-1}p_{i+1} \leq p_i^2$.

Monotone hazard rate (MHR) distributions over $[n]$ (denoted by $\mathcal{MHR}_n$), are distributions $p$ with CDF $F$ for which $i < j$ implies $\frac{f_i}{1-F_i} \le \frac{f_j}{1-F_j}$.

The following theorem bounds the complexity of testing these classes (for moderate $\varepsilon$).

**Theorem 4.** *Suppose $\varepsilon > n^{-1/5}$. For each of the classes, unimodal, log-concave, and MHR, there exists an algorithm for testing the class over $[n]$ with sample complexity $O(\sqrt{n}/\varepsilon^2)$.*

This result is a corollary of the specific results for each class, which is proved in the appendix. In particular, a more complete statement for unimodality, log-concavity and monotone-hazard rate, with precise dependence on both $n$ and $\varepsilon$ is given in Theorems 7, 8 and 9 respectively. We mention some key points about each class, and refer the reader to the respective appendix for further details.

**Testing Unimodality** Using a union bound argument, one can use the results on testing monotonicity to give an algorithm with $O\left(\sqrt{n}\log n/\varepsilon^2\right)$ samples. However, this is unsatisfactory, since our lower bound (and as we will demonstrate, the true complexity of this problem) is $\sqrt{n}/\varepsilon^2$. We overcome the logarithmic barrier introduced by the union bound, by employing a non-oblivious decomposition of the domain, and using Kolmogorov's max-inequality.

**Testing Log-Concavity** The key step is to design an algorithm to learn a log-concave distribution in $\chi^2$ distance. We formulate the problem as a linear program in the logarithms of the distribution and show that using $O(1/\varepsilon^5)$ samples, it is possible to output a log-concave distribution that has a $\chi^2$ distance at most $O(\varepsilon^2)$ from the underlying log-concave distribution.

**Testing Monotone Hazard Rate** For learning MHR distributions in $\chi^2$ distance, we formulate a linear program in the logarithms of the CDF and show that using $O(\log(n/\varepsilon)/\varepsilon^5)$ samples, it is possible to output a MHR distribution that has a $\chi^2$ distance at most $O(\varepsilon^2)$ from the underlying MHR distribution.

# 8 Lower Bounds

We now prove sharp lower bounds for the classes of distributions we consider. We show that the example studied by Paninski [10] to prove lower bounds on testing uniformity can be used to prove lower bounds for the classes we consider. They consider a class $\mathcal{Q}$ consisting of $2^{n/2}$ distributions defined as follows. Without loss of generality assume that $n$ is even. For each of the $2^{n/2}$ vectors $z_0 z_1 \ldots z_{n/2-1} \in \{-1, 1\}^{n/2}$, define a distribution $q \in \mathcal{Q}$ over $[n]$ as follows.

$$q_i = \begin{cases} \frac{(1+z_\ell c\varepsilon)}{n} & \text{for } i = 2\ell + 1 \\ \frac{(1-z_\ell c\varepsilon)}{n} & \text{for } i = 2\ell. \end{cases} \tag{2}$$

Each distribution in $\mathcal{Q}$ has a total variation distance $c\varepsilon/2$ from $U_n$, the uniform distribution over $[n]$. By choosing $c$ to be an appropriate constant, Paninski [10] showed that a distribution picked uniformly at random from $\mathcal{Q}$ cannot be distinguished from $U_n$ with fewer than $\sqrt{n}/\varepsilon^2$ samples with probability at least $2/3$.

Suppose $\mathcal{C}$ is a class of distributions such that (i) The uniform distribution $U_n$ is in $\mathcal{C}$, (ii) For appropriately chosen $c$, $d_{\mathrm{TV}}(\mathcal{C}, \mathcal{Q}) \ge \varepsilon$, then testing $\mathcal{C}$ is not easier than distinguishing $U_n$ from $\mathcal{Q}$. Invoking [10] immediately implies that testing the class $\mathcal{C}$ requires $\Omega(\sqrt{n}/\varepsilon^2)$ samples.

The lower bounds for all the one dimensional distributions will follow directly from this construction, and for testing monotonicity in higher dimensions, we extend this construction to $d \ge 1$, appropriately. These arguments are proved in Section H, leading to the following lower bounds for testing these classes:

**Theorem 5.**

- *For any $d \ge 1$, any algorithm for testing monotonicity over $[n]^d$ requires $\Omega(n^{d/2}/\varepsilon^2)$ samples.*

- *For $d \ge 1$, testing independence over $[n_1] \times \cdots \times [n_d]$ requires $\Omega\left((n_1 n_2 \ldots n_d)^{1/2}/\varepsilon^2\right)$ samples.*

- *Testing unimodality, log-concavity, or monotone hazard rate over $[n]$ needs $\Omega(\sqrt{n}/\varepsilon^2)$ samples.*

## Footnotes

[1] We want success probability at least $2/3$, which can be boosted to $1 - \delta$ by repeating the test $O(\log(1/\delta))$ times and taking the majority.

[2]We also require the algorithm to output a description of an effective support for which this property holds. This requirement can be slightly relaxed, as we show in our results for testing unimodality.

[3]This definition describes monotone non-increasing distributions. By symmetry, identical results hold for monotone non-decreasing distributions.

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
