[Reviews · NeurIPS 2015]

Submitted by Assigned_Reviewer_1

This paper has results about tests like:

H0:

P is monotonone

versus

H1: P is no monotone

I like the paper.

The results are interesting. However, I have a few concerns:

1. The sample space you use is a discrete cube.

Why would anyone want

to test for monotonicity over this space?

In practice, people use

monotinicity tests for continuous random variables.

Can you cite

any real data analysis problem where your setting is of scientific interest?

2. You dismiss the statistics literature as concentrating on the large

sample regime.

This is a bit misleading: many of the tests can be

made exact by simulating the null distribution. The use of asymptotics

is often to get precise theoretical results about the power of the

tests. By precise I mean limits, not bounds.

And there are

statistics papers that do provide finite sample guarantees.

An

example is:

Dumbgen, Lutz, and Guenther Walther.

"Multiscale inference about a density." The Annals of Statistics (2008): 1758-1785.

At any rate, your references to the vast statistical literature on this topic

is too sparse. You need to have more references to the statistical

literature.

3. Also, I am not convinced your test really is a finite sample test.

Suppose I want to use your test and I want to make sure the type I

error is less than alpha. (You take alpha = 1/3 but I assume you

can change things to make alpha any user-specified level.) Your

results say: there exists some N0 such that the type I error is

less than alpha if the sample size N is larger than N, the type I

error is less than alpha. The asymptotic statistics tests say: for

large N the test has type I error close to alpha. I don't see any

real difference. If I use your test with real data, I have no way

of knowing if N is bigger than N0. We simply have to assume N is

large enough. So it seems to me that, at a practical level the same

as an asymptotic test.

Summary: Interesting paper. But I am not convinced this problem actually comes up in practice. Also, the connections to statistics are dismissed too readily.

Submitted by Assigned_Reviewer_2

I only give a very quick pass on the paper because it is supposed to be a "light" review. I found the materials interesting and very well motivated.

Summary: The paper looks interesting.

Submitted by Assigned_Reviewer_3

It seems to be perfect (I did not understand the details, to be honest). I am wondering why the tester is good only for monotonic, log-concave, unimodal, and monotonic hazard rate. It would be pleasing to explain why.
Summary: Considering a novel use of the chi square statistic, the paper constructs a sample-optimal and computationally efficient testers for class C and epsilon.

Submitted by Assigned_Reviewer_4

The problem of testing membership of an unknown distribution to the sets of monotone, log-concave, unimodal, and monotone hazard rate distributions over [n] (or [n]^d, for monotonicity) is considered. The approach to each problem follows a nice general framework -- first, an estimate q of the unknown distribution p is constructed, which effectively assumes that p belongs to the class being considered to reduce sample complexity. A modified chi-squared test is then applied to test whether p and q are close. The estimators necessary to achieve the first step are novel and unique to each class of distributions considered.

This two step procedure avoids the need for obvious alternatives such as testing equality between p and, say, a net over the class in question, which incurs large computational and statistical costs. The proposed approach gives efficient algorithms with improved (and optimal) sample complexities.

It is perhaps somewhat noteworthy that not only are the (optimal) rates for each of the four classes identical (in the case d=1, for monotone distributions), but that the construction used in the matching lower bound for each problem is identical.

Including some definitions would improve clarity for readers not familiar with this particular subject, e.g. the partial order in the definition of monotone distributions over the hypergrid. (Also, line 161 promises definitions of the distances in the appendix, which seem to be missing.)
Summary: The paper makes significant contributions to testing shape-constrained discrete distributions.

Submitted by Assigned_Reviewer_5

This paper derives a new method and new bounds for testing whether a distribution belongs to a certain class (eg monotone, log-concave, uni-modal) distribution. The authors provide both upper and lower bounds for the classes they consider (showing that their upper bounds are essentially tight). They also provide some simulations on synthetic data for their algorithm.

The paper is generally well written and the results seem correct. The authors also provide adequate comparisons with earlier bounds and discuss carefully in which regimes their result provides a better rate.

My main concern about this work is that it only considers discrete distributions. It would be nice if the authors could provide some motivational discussion on whether the actual testing problems they consider occur in practice. Further it would be nice to see some discussion on whether the results are extendable to non-discrete distribution (maybe under sone assumptions on the density?).

Minor comments: The authors could make a stronger effort to make it a self contained read. Eg, on page 3 they refer the reader to the appendix for some basic definitions, but those definitions are not in the appendix.

Summary: This paper derives a new method and new bounds for testing whether a distribution belongs to a certain class of (discrete distrbutions). The results are well presented.

Submitted by Assigned_Reviewer_6

This manuscript aims to the determine the number of observations required to distinguish an unknown distribution $p$ from some class of distributions $\mathcal{C}$ that is at least $\epsilon$ away in total variation distance. Different classes of $\mathcal{C}$ considered include monotone, log-concave, unimodal and moonotone hazard rate distributions.

Below are a few questions and comments which will hopefully help the author(s) improve the manuscript:

A. Motivations I feel that the motivations for imposing shape-related constraints on discrete distributions remain to be explained further in this manuscript. Discrete distributions are mainly useful for modelling (i) the frequencies of observed objects (e.g. IP addresses) and (ii) counting data. In (i), applying monotone constraint means that one has to impose an ordinal structure on the objects; furthermore, one needs a metric associated with the ordinal structure to use the log-concave constraint. These implications somewhat limit the use of the proposed method, since not all the objects have a natural order.

B. Theory. B1. Proof of Theorem 1.

In order to have (1) and (2), one needs independence between the observations $X_i$ and the newly-drawn distribution $q$ (which satisfies Properties 1 and 2). This is not always the case, unless the observations used to estimate $q$ are discarded for the rest of the analysis. This requirement should be stated more explicitly.

B2. Use of Lemma 5.

In the monotone case, Lemma 5 states that one can find $q$ such that $\mathbb{E}[\chi^2(p,q)]\le \epsilon^2/500$. However, in order to invoke Theorem~1, one requires $\chi^2(p,q) \le \epsilon^2/500$. So there is a gap in between.

B3. Continuous case vs discrete case.

Birges (1987) was cited a few times in the development of the theory. To my knowledge, the work of Birges only deals with density estimation. Could the author(s) state the particular result from Birges (1987) that has been referred to many times in the manuscript?

B4. Log-concavity vs unimodality.

In the proof of Lemma 7, as well as in Section H.2 in the appendix, it is stated that `any log-concave distribution is unimodal'. However, this is false in view of the definition given in Section~2 (i.e. $f_{i-1}f_{i+1} \le f_i^2$). For example, consider $n=7$, $f_1 = f_4 = f_7 = 1/3$ and $f_2 = f_3 = f_5 = f_6 = 0$.

B5. Rate for testing log-concavity.

It is stated in the abstract that testing log-concavity requires $O(sqrt{n}/\epsilon^2)$ samples. However, in view of Theorem 4, this is true when $\epsilon > n^{-1/4}$. Clearly, this statement is not precise when we are in the region of `fixing $n$ and decreasing $\epsilon$'.

C. Connections to Statistics. It is worth pointing out that here the author(s) assumed that the distribution of interest, i.e. $p$, is discrete. In addition, in the development of theory, the support of $p$ is assumed to be $[n]$ (i.e., $\{1,2,\ldots,n\}$), which grows with respect to $n$. This setting is quite popular in Computer Science, though perhaps is not very well-known in Statistics. The author(s) cited work such as Hall and Van Keilegom (2005) and Cule and Samworth (2010). However, the cited work only deal with the continuous case. For more relevant work on estimating/testing a discrete monotone and log-concave distribution, see Jankowski and Wellner(2009) and Balabdaoui et.al.(2013).

References

Jankowski, H. and Wellner, J.A. (2009). Estimation of a discrete monotone distribution. Electronic Journal of Statistics, 3, 1567--1605.

Balabdaoui, F., Jankowski, H., Rufibach, K. and Pavlides, M. (2013). Asymptotic distribution of the discrete log-concave MLE and related applications, Journal of Royal Statistical Society Series B, 75, 769-790.

Summary: The main method illustrated in this manuscript is built upon Valiant and Valiant (2014), though the new settings are substantially different, which lead to the development of new theory (as well as new lemmas in the appendix).

I feel that the main contribution of this manuscript is its theory (which seems interesting), though I remain to be convinced that the proofs are rigorous enough.

Author Feedback
Author rebuttal: We thank the reviewers for the thoughtful comments and feedback, and believe they
will make the paper better. We first address comments which are common to multiple reviewers and then answer the reviewers individually.

Property testing on discrete and continuous supports:
Our paper was primarily motivated by the long line of research in property testing and sublinear algorithms. Assuming discrete distributions makes for a very clean formulation and we can study the tradeoffs between the support size n and sample complexity. We settle the line of research with respect to the testing problems we consider, including monotonicity. We also provide optimal testers for independence, a result to be included in the final version.

Discrete objects are interesting in their own right. Most of the data we observe is already discretized. For example, we can look at the sales of cars with respect to age and income, and in most cases the observations are already quantized, providing a natural discrete ordering. We also believe our results can extend to study monotonicity and unimodality over other discrete objects, including graphs.

This being said, we agree that continuous distributions are very interesting, and must be investigated further. It is well known that further restrictions on the distributions or the metric are needed to obtain non-trivial results in this setting. In preliminary follow-up work, we extend our result to testing monotonicity of continuous
distributions with respect to Wasserstein distance.

Statistical literature:
We agree that there has been prior work in statistics in the sample-regime that we consider. Indeed, it is hard to do justice to the vast literature in limited space. In the final version we will do a more thorough job of comparing to the relevant statistics literature, e.g.,

D. Huang, S. Meyn "Generalized Error Exponents for Small Sample Universal Hypothesis Testing".

Specific comments:
Reviewer_1: Questions 1 and 2 are addressed above.
Question 3: Our paper does provide specific bounds on the value of N0. In particular,
if the number of samples N is at least c*n^(1/2)/eps^2 (for a constant c that we can upper bound), we know that our algorithms work.

Reviewer_2: The exclusion of the definitions was a compilation error on our part, and they are now correctly included in the current draft.

Reviewer_3: Please refer to the discussion at the beginning about continuous and discrete distributions. The exclusion of the definitions was a compilation error on our part, and they are now correctly included in the current draft.

Reviewer_4: Indeed, our results are applicable to many more classes than those mentioned in the submitted draft, such as optimal testing independence of random variables (which we have included in the current draft). More generally, our method applies for any class of distributions which can be highly efficiently learned in chi^2 distance.

Reviewer_5: We thank the reviewer for the encouraging comments.

Reviewer_6: Please refer to the discussion in the beginning.

Reviewer_7:
A. Please see the discussion of discrete and continuous supports in the beginning.
B. Apologies for being unclear in these parts of our paper, we address the individual
points below, and will be more explicit on all of these in the full version.
B1. Indeed, the algorithm does what you suggest -- we split the sample into two parts, one of which is used for the learning stage, and one is used for the testing
stage.
B2. We meant to explicitly make the constant for expectation of chi-squared distance smaller and state an application of Markov's inequality, thus guaranteeing the requirements of our tester.
B3. Indeed, Birge's result is stated for continuous monotone densities. However it
extends to the discrete case, with H*W replaced with 'n' in the results, and the same proof still holds. For more discussion of this, please see Section 1.2 and Appendix A of the paper "Learning k-Modal Distributions via Testing" by Daskalakis, Diakonikolas, and Servedio.
B4. This is an oversight, we should also state the restriction that the support is contiguous. Under the current definition, log concave distributions would be too general -- any arbitrary distribution which is supported on the numbers equivalent to 0 (mod 3) would be log concave. Please see Definition 2.1 in "Asymptotics of the discrete log-concave maximum likelihood estimator and related applications" by Balabdouai, Jankowski, Rufibach, Pavlides. This reference was missing due to the space-constraints, but we will do better in the final version to fit in more of
these relevant citations.
B5. We explicitly state the restriction of large n vs. 1/eps in the theorem statements.
We will do so in the abstract (i.e., mention that the focus is when eps > 1/n^{O(1)}).
C. Thank you for the references, which are now included in the current draft.